# Color Stability and Staining Susceptibility of Direct Resin-Based Composites after Light-Activated In-Office Bleaching

**DOI:** 10.3390/polym13172941

**Published:** 2021-08-31

**Authors:** Pei-Wen Peng, Chiung-Fang Huang, Ching-Ying Hsu, Ann Chen, Ho-Him Ng, Man-Si Cheng, Shiang Tsay, Jia-Yi Lai, Tzu-Sen Yang, Wei-Fang Lee

**Affiliations:** 1School of Dental Technology, Taipei Medical University, Taipei 11031, Taiwan; apon@tmu.edu.tw (P.-W.P.); chiung0102@tmu.edu.tw (C.-F.H.); 2Division of Family and Operative Dentistry, Department of Dentistry, Taipei Medical University Hospital, Taipei 11031, Taiwan; 3Department of Dentistry, Taipei Medical University Hospital, Taipei 11031, Taiwan; r590814@gmail.com; 4School of Dentistry, Taipei Medical University, Taipei 11031, Taiwan; b202105074@tmu.edu.tw (A.C.); b202105077@tmu.edu.tw (H.-H.N.); b202105072@tmu.edu.tw (M.-S.C.); b202105069@tmu.edu.tw (S.T.); b202105041@tmu.edu.tw (J.-Y.L.); 5Graduate Institute of Biomedical Optomechatronics, Taipei Medical University, Taipei 11031, Taiwan

**Keywords:** red wine, bleaching, resin-based composites, color, whiteness index

## Abstract

This study evaluated color stability and staining susceptibility of five direct resin-based composites (RBCs) subjected to light-activated in-office bleaching with 40% hydrogen peroxide (HP). The test materials included 5 RBCs, which consisted of one nano-filled, one sub-micron, one bulk-filled, and two nano-hybrid RBC types. Ten disc-shaped specimens of each RBC were fabricated and divided into bleaching (BLE) and non-bleaching (CON) groups (*n* = 5 for each group). Specimens were then immersed in red wine solution over 4 h. A spectrophotometer was used to obtain Commission Internationale de l’Eclairage (CIE) L*a*b* parameters for each of the following periods tested: before bleaching (T_BA_), after bleaching (T_BL_), and after staining (T_ST_). Color stability and staining susceptibility were evaluated using two metrics, CIEDE2000 color differences (ΔE_00_) and whiteness variations using the whiteness index (ΔWI_D_). Data were analyzed using repeated measures two-way analysis of variance (ANOVA) (α = 0.05). Statistically significant and clinically unaccepted ΔE_00_ and ΔWI_D_ were observed for all tested specimens between T_BA_ and T_BL_. The nano-hybrid type RBCs showed the highest discoloration among materials after bleaching treatment. The BLE group exhibited significantly higher ΔE_00_ and ΔWI_D_ than the CON group for all the tested RBCs between T_BA_ and T_ST_. The sub-micron type RBC showed the highest discoloration among materials after immersion in the red wine. Conclusion. The light-activated in-office bleaching with 40% HP’s influences on color and whiteness index were material-dependent. The use of bleaching treatment also increased the susceptibility to red wine for all RBCs.

## 1. Introduction

Recently, bulk-fill and nano-filled or nano-hybrid resin-based composites (RBCs) have been introduced for direct restorations and are considered the most popular tooth-colored filling materials in dental restorations due to their natural appearance, low costs, and longevity [1,2]. Although tooth-colored restorations should provide excellent color match during clinical service and high color stability over a long period in an oral environment, RBCs are prone to chromatic alteration caused by either intrinsic or extrinsic factors. Pigments in food or beverages are the extrinsic factors that cause a color mismatch of these materials, resulting in treatment failure due to the lack of the esthetic zone [3,4]. Studies demonstrated that adsorption or absorption of beverages with intense dark colors stain the RBCs severely, resulting in a clinically visible color difference [5,6]. Red wine, a commonly consumed drink worldwide, is also a staining beverage that caused the clinically unacceptable mismatch of color difference (ΔE_00_) > 1.79 for various direct composite resins, including nanocomposites, giomers, and resin-modified glass ionomers using CIEDE2000 color difference formulas (Commission Internationale de l’Eclairage, CIE) [7,8,9,10]. It was also reported that different types of materials exhibited different levels of resistance to discoloration [11]. Therefore, it is worthwhile to improve the capacity of these esthetic materials to resist discoloration and regain the overall whiteness.

Whitening is a useful strategy to treat restorative materials effectively with an increased desire for aesthetics by changing the intrinsic color or removing extrinsic stains [12]. The use of bleaching agents is a constantly developed procedure for whitening the stained restorative materials. The whiteness index (WI_D_) was recently introduced to interpret the level of white under both laboratory and clinical conditions with high WI_D_ indicating a white specimen [13].

Hydrogen peroxide (HP), with concentrations varying from 3% to 40%, decomposed into hydroxy-free radicals under light or heat irritations, is the most commonly used bleaching agent to dissociate double bonds or ring structures present within stains [14,15,16]. The bleaching efficacy is typically attributed to the bleaching protocols, including activation sources, HP concentration, duration of the bleaching, and the structures of restorative materials, such as the structure of the resin matrix as well as the characteristics of the filler particles [17,18,19]. The higher the HP concentration, the greater the de-pigmentation process, resulting in stain removal, and the color changes of RBCs [20]. Although the effects of HP on the color change of RBCs remain contentious, it is generally agreed that different types of resin-based composites reveal different resistances to bleaching [21,22,23,24].

The consequences of bleaching may result in alterations in microhardness and roughness [25]. An increase in superficial roughness was predisposed to biofilm formation and extrinsic staining. However, studies comparing and assessing color stability of current RBCs using an in-office whitening product and the capacity to resist staining by red wine are limited [19]. Therefore, the purpose of the present study was to evaluate the color stability and staining susceptibility of five RBCs after light-activated in-office bleaching with 40% HP. The null hypothesis was that a 40% HP bleaching agent would not alter the color and whiteness index of RBCs. In addition, there would be no significant difference regarding red wine stain-resistances among these materials, irrespective of light-activated in-office bleaching with 40% HP.

## 2. Materials and Methods

Five resin-based composites, one staining solution, and one in-office whitening product evaluated in the present study are shown in Table 1. Figure 1 describes the study design. Ten disc-shaped specimens of each RBC were fabricated and divided into two groups of five samples each. The BLE group was subjected to the light-activated in-office bleaching with 40% HP, while the CON group received no bleaching treatment.

### 2.1. Specimen Preparation

Disc-shaped (10.5 mm × 1.6 mm) specimens (*n* = 10) were fabricated from each material using a stainless-steel screw washer mold. Each composite was packed into the mold, which was positioned over a polyester strip on a glass plate, and gently pressed with a transparent acrylic sheet on the top of the mold. The specimens were then light-cured with a light-emitting diode (LED) device with approximately 800 mW/cm^2^ for 80 s (3M ESPE Dental Products; Monrovia, CA, USA) to ensure full conversion of the photoactivation. Immediately, the specimens were polished using SiC #1000 sandpaper and diamond paste for 20 s, respectively. After rinsing and gently drying, the thickness of the specimens was then verified with a digital micrometer (MDC-250; Mitutoyo, Kawasaki, Japan), and stored in deionized water at 37 °C for 24 h.

### 2.2. Bleaching Procedure

Specimens from the bleaching group (*n* = 5 in each material) were subjected to the light-activated in-office bleaching containing 40% HP (Table 1) by using an LED device (3M ESPE Dental Products; Monrovia, CA, USA) according to the manufacturer recommendations. The whitening gel was painted in a 2-mm-thick layer on the surface at room temperature and cured for 30 s 15 times with a 10-s interval. Three bleaching sessions were performed with a 1-h break. After each bleaching session, specimens were cleaned with distilled water for 30 s and stored in distilled water until the next bleaching session.

### 2.3. Staining Procedure

Each specimen was placed in a container with 10 mL of red wine (Cabernet Sauvignon) at room temperature for 3 h. After rinsing with distilled water for 10 s, specimens were brushed for 1 min under running water using an electric toothbrush (MP-DH200-BL, Maruman Pro Sonic 2, Tokyo, Japan).

### 2.4. Color Measurements

The color of each specimen was measured in 3 different measuring times: before bleaching (denoted as the baseline, T_BA_), after bleaching (T_BL_), and after staining (T_ST_). The color of each specimen was measured in triplicate and expressed by CIELAB system using a spectrophotometer (Easyshade; VITA Zahnfabrik, Bad Säckingen, Germany). In this system, L* represents the lightness, ranging between 0 (dark) and 100 (bright), a* represents the red-green chromaticity coordinate, and b* represents the yellow-blue chromaticity coordinate [26,27]. The chroma (C) and hue (H) of the specimen were also numerically obtained.

After carefully drying with tissue papers, the color parameters were obtained against a black background. The CIEDE2000 color difference (ΔE_00_) was calculated between two different timepoints using the following formula:$$\Delta E_{00}=\sqrt{{\left(\frac{L_{i}-L_{j}}{K_{L}S_{L}}\right)}^{2}+{\left(\frac{C_{i}-C_{j}}{K_{C}S_{C}}\right)}^{2}+{\left(\frac{H_{i}-H_{j}}{K_{H}S_{H}}\right)}^{2}+R_{T}\left(\frac{C_{i}-C_{j}}{K_{C}S_{C}}\right)\left(\frac{H_{i}-H_{j}}{K_{H}S_{H}}\right)}$$
where the subscripts i and j refer to the specimen measured at two different timepoints; S_L_, S_C_, and S_H_ represent weighting functions; K_L_, K_C_, and K_H_ are parametric factors, which were set to 1 in the present study; and R_T_ is the rotation function.

The 50:50% perceptibility (PT) and 50:50% acceptability (AT) thresholds in the present study were set at ΔE_00_ of 0.8 and 1.8, respectively [8].

Moreover, the degree of whiteness (WI_D_) was quantified as follows [28]:$$WI_{D}=0.55L^{*}-2.32a^{*}-1.100b^{*}$$

The difference in whiteness index between two different time points (ΔWI_D_) was calculated. The 50:50% whiteness perceptibility (WPT) and whiteness acceptability (WAT) thresholds were considered as ΔWI_D_ of 0.61 and 2.90, respectively [29].

### 2.5. Evaluation of Surface Roughness

The surface roughness was measured using a portable surface roughness tester (TR200; SaluTron Messtechnik GmbH, Frechen, Germany). Each of a total of five measurements was obtained using a cutoff of 0.25 mm, and the arithmetic mean (Ra) was calculated of the measurements taken before (T_BA_) and after (T_BL_) the bleaching procedures. One specimen from each group was selected to observe the surface morphology at T_BA_ and T_BL_ using scanning electron microscopy (SEM; TM4000, Hitachi, Tokyo, Japan).

### 2.6. Statistical Analyses

A statistical software program (IBM SPSS Statistics, v19.0; IBM Corp) was used for data analysis. Data from color parameters (CIE L*, a* and b*), WI_D_, ΔE_00_, and ΔWI_D_ were statistically analyzed using the one-way analysis of variance (ANOVA). A post hoc Tukey’s multiple comparison test (α = 0.05) was used to identify material differences. Data from surface roughness, ΔE_00_, and ΔWI_D_ were also statistically analyzed using Student’s t-test to compare differences between the BLE and CON groups.

## 3. Results

Figure 1 presents surface roughness measurements of materials in BLE groups at TBA and TBL. Surface roughness increased after bleaching for all materials and showed significant alterations for all materials (*p* = 0.004 for Z350; *p* = 0.006 for GRA; *p* = 0.011 for BFP; *p* < 0.001 for ESQ and OPT). After bleaching, OPT presented the highest Ra values, whereas GRA showed the lowest surface roughness (Ra) values among groups.

Figure 2 shows SEM images of BLE groups at T_BA_ and T_BL_, confirming the results of the surface roughness measurement. Surface alterations were observed after bleaching procedures, such as the less uniform surfaces, dislodged particles, and protruding filler particles.

Table 2 shows the mean and standard deviation of color parameters in CIELab systems and the WI_D_ of each material from the BLE group at three different time points. After bleaching, CIE L* and WI_D_ values decreased for ESQ and Z350, whereas those for GRA, BFP, and OPT increased. The staining procedure produced the lowest CIE L* and WI_D_ values among three-time points for all materials. The CIE a* values of ESQ and Z350 increased, whereas those for GRA, BFP, and OPT decreased after bleaching. The bleaching procedure decreased the CIE b* value, depicting a shift toward blue for all materials. After staining, the color for all materials shifted from green and blue toward red and yellow.

Figure 3 shows the mean and standard deviation of color difference (ΔE_00_) between different measuring time points of each material from the BLE and CON groups. One-way ANOVA results revealed significant color differences among materials after immersion in red wine in the CON group (*p* < 0.001). Z350 showed the highest ΔE_00_ among all the materials (*p* < 0.001 for all materials), followed by BFP, ESQ, GRA, and OPT. However, there were no statistical differences among other materials. All materials showed ΔE_00_ values above AT.

In the BLE group, all materials showed ΔE_00_ values above AT after bleaching treatment. GRA, BLE, and OPT (*p* = 0.492 for GRA and BLE; *p* = 0.79 for OPT and BLE; *p* = 0.079 for GRA and OPT) revealed statistically higher ΔE_00_ values than ESQ and OPT (*p* = 0.364). After the staining procedure, we noted that the BLE group yielded significantly greater ΔE_00_ values than the CON group for all materials (*p* < 0.001 for materials) except for BFP (*p* = 0.082).

Figure 4 shows the mean and standard deviation of difference whiteness index (ΔWI_D_) between different measuring time points of each material from the BLE and the CON groups. In the CON group, all materials became darker and showed ΔWI_D_ values above WAT after immersion in red wine. After bleaching, there were significant differences among materials (*p* < 0.001) in the BLE group. GRA, BFP, and OPT appeared brighter. All materials showed ΔWI_D_ values above WAT after bleaching, except for Z350, with the following decreasing order: GRA = OPT (*p* = 0.975) > BFP (*p* < 0.001 with ESQ and Z350; *p* = 0.043 with GRA; *p* = 0.012 with OPT) > ESQ (*p* < 0.001 with all materials) > Z350 (*p* < 0.001 with all materials). After the staining procedure, it was also noted that the BLE group revealed significantly greater ΔWI_D_ values than the CON group for all materials (*p* < 0.001 for ESQ; *p* = 0.001 for Z350 and GRA; *p* = 0.002 for BFP; *p* = 0.009 for OPT).

## 4. Discussion

The present study evaluated the influence of the light-activated bleaching treatment with 40% HP on color difference, whiteness variation, and stain susceptibility of the resin-based composites. According to the study results, the first hypothesis was rejected because a light-activated bleaching treatment with 40% HP has a significant effect on the color and whiteness of the tested resin-based composites. The second hypothesis was similarly rejected because specimens with a light-activated bleaching treatment with 40%HP exhibited highly red wine staining-susceptibility compared to those without the bleaching treatment.

In the present study, color measurements were carried out on a black background to reflect the real oral environment while evaluating the specimens’ final colors. Two metrics (ΔE_00_ and ΔWI_D_) were used to evaluate the bleaching effect on the color of resin-based composites. The acceptability and perceptibility thresholds for the ΔE_00_ value, although the values were still not conclusive in studies, they gave the clinical meanings, the color similarity or change [8]. The ΔWI_D_ value, recently published by studies [28,29], provided sufficient information about whiter or darker changes of a specimen after bleaching [16].

The color stain resistance of resin-based restorations in the oral environment is an essential requirement against saliva, food, and drink, which are the common extrinsic factors causing the discoloration of the restorative materials [4]. Red wine has been reported to cause significant staining in hybrid resins and glass-ceramics [10]. Therefore, the effect of immersion in red wine, containing red and blue colorants, is a plausible procedure to evaluate the staining tendency in resin-based materials. For CON groups, all materials showed ΔE_00_ values above AT (Figure 1), which aligned with previous findings [10].

According to the information provided from the manufacturers, four types of RBCs, one sub-micron-filled (ESQ), one nano-filled (Z350), one bulk-filled (BFP), and two nano-hybrid (OPT and GRA), classified by filler-size distribution, were used in the present study, which were different in the chemical structure of the organic matrix as well as the percentage and size of inorganic fillers (Table 1). The total percentage of weight of inorganic fillers evaluated in the present study decreases as follows: GRA (87%) > ESQ (82%) > OPT (79.5%) > BFP (76.5%) > Z350 (72.5%). The filler particle type significantly affected the color stability of the resin-based composites after 4 h of immersion in red wine (Figure 1). The type of resin matrix, such as Bisphenol A-Glycidyl Methacrylate (Bis-GMA) or Urethane Dimethacrylate (UDMA) had a significant role in the staining susceptibility. The nano-hybrid type (OPT and GRA) exhibited the highest color stability, followed by sub-micron-filled, bulk-filled, and nano-filled types, which might be attributed to resin matrix volume content, the absence of UDMA, and the presence of the quantum effect [9,19].

The bleaching treatments, including both at-home and in-office products, have been demonstrated to improve stain removal from the resin composites effectively but caused color changes [7]. Pecho et al. [23] showed that tested composites, including nanohybrid-, microhybrid-, and microfilled-types, showed clinically acceptable after bleaching with 35% HP when the visual threshold was 2.7. The present study showed color differences of all tested composites were above the visual threshold after bleaching with 40% HP, especially for nanohybrid- and bulk-filled types. These controversial results may be attributed to the light source in the bleaching gel activation, which was reported to improve whitening capacity and bleaching efficacy [17]. Further in-depth studies will be needed to clarify this presumption.

While the nano-filled and sub-micron type specimens become dark after the bleaching procedure, the nano-hybrid and bulk-filled types specimens were brighter after bleaching (Table 2), which is consistent with previous findings [16,24]. A similar trend was observed in the whiteness variations (Figure 2). The number of color differences of RBCs in the present study after bleaching appeared related to the amount of filler and the filler type and the difference in organic matrix structure [16]. The nano-hybrid and bulk-filled types RBCs showed similar color differences, which were higher than those of nano-filled and sub-micron types.

In agreement with a previous study [25], the light-activated bleaching treatment with 40%HP produced rougher surfaces of the RBCs. The influences of bleaching treatment on color changes in RCBs were caused by multi factors, including the composites tested, the bleaching agent used, and their interactions [18,19,23]. The present study confirmed that the BLE group presented higher ΔE_00_ values and lower ΔWI_D_ values after red wine staining than the CON group for all tested materials, which corroborated the considerations above, and is consistent with a previous finding [19], and could be due to the effect of the bleaching agents on the RBCs. For instance, contact with the bleaching agents could result in roughness alternations related to the superficial softening and chemical degradation of the RCB’s matrices. The nano-filled type RBC exhibited bleaching resistance similar to that seen with the sub-micro-filled type one, but results after staining differed. 

The monomer conversion ratios in the present study for all materials were not determined, but a long photoactivation time was applied to provide the high conversion. Hydrogen peroxide is an aggressive oxidant, resulting in the elution of unpolymerized monomers and unspecific oxidative products from composites [23]. It was reported the interactions between resin-based composites with different degrees of polymerization and the application of bleaching agents [16]; however, they need to be confirmed in further studies. The present study did not truly replicate the intra-oral conditions, which were also considered limitations to this study. The thermal stress resulting from thermocycling should be further investigated. Another limitation was that the specimens were stored in deionized water. In our further study, artificial saliva would be used.

## 5. Conclusions

Within the limitations of the present study, the light-activated bleaching treatment with 40% HP produced unacceptable color changes and whiteness variations of the tested resin-based composites. Surface alterations and increased roughness were also observed after bleaching procedures for all resin-based composites. These effects were material-dependent, and the bleached RBCs increased the susceptibility to red wine for all RBCs.

## Figures and Tables

**Figure 1 polymers-13-02941-f001:**
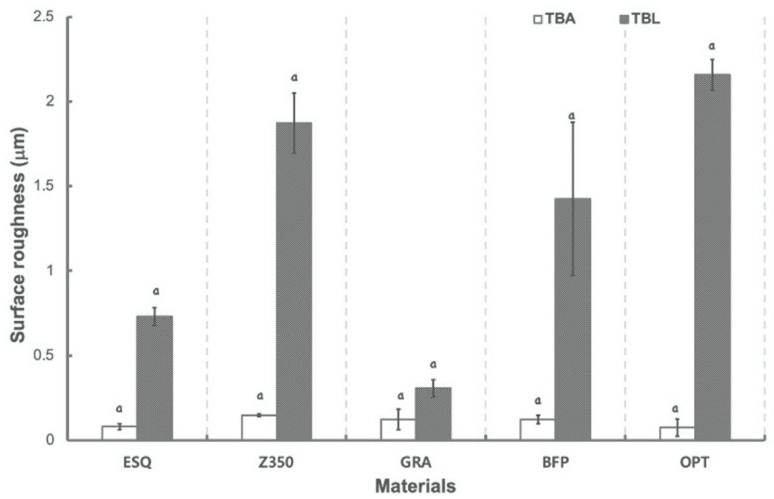
Mean and standard deviation values of surface roughness for each material before (TBA) and after (TBL) bleaching. Same lowercase letter indicates a static difference between time points for each material.

**Figure 2 polymers-13-02941-f002:**
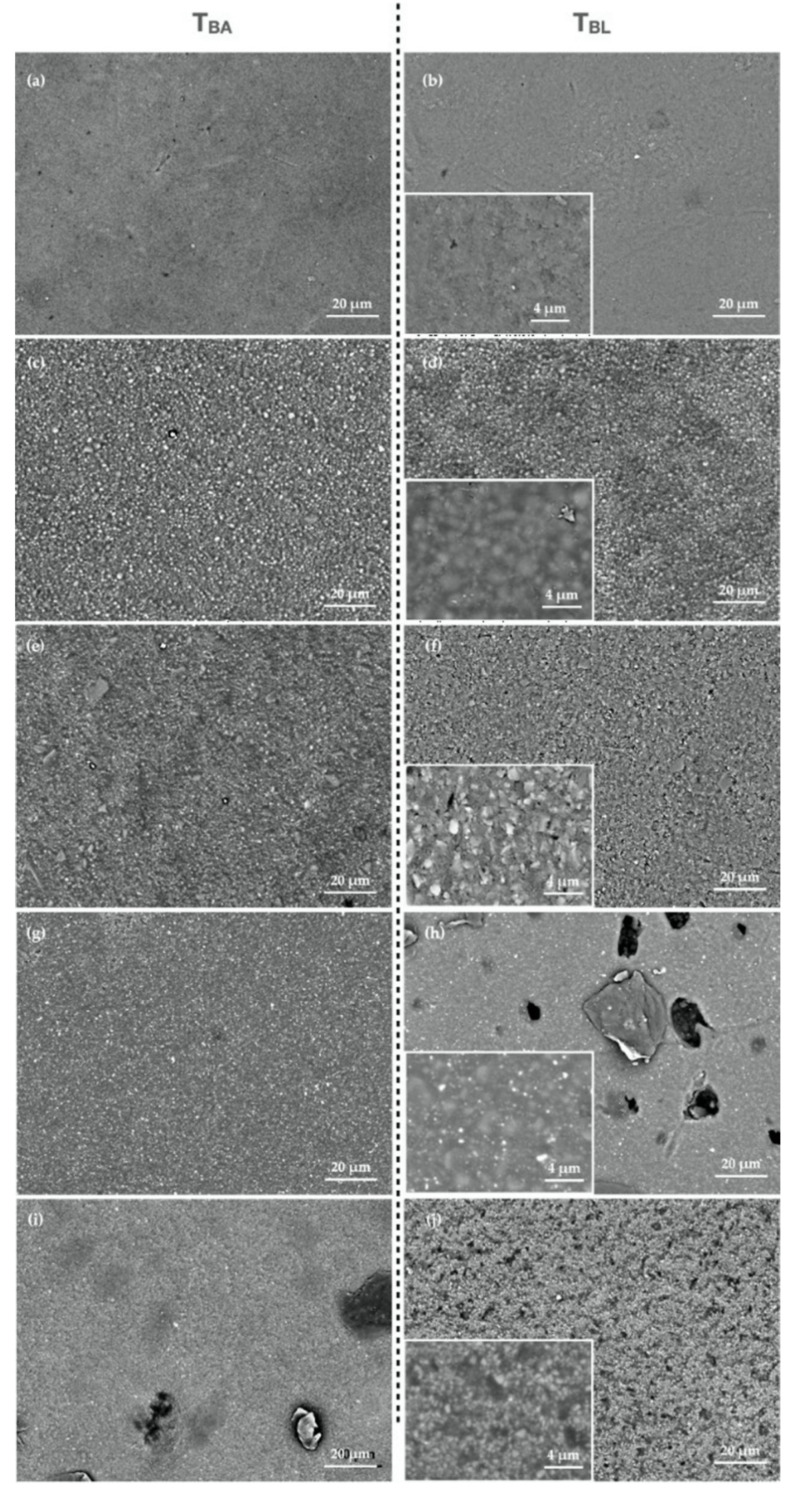
Scanning electron microscope images of (**a**,**b**) ESQ, (**c**,**d**) Z350, (**e**,**f**) GRA, (**g**,**h**) BFP, and (**i**,**j**) OPT before (T_BA_) and after (T_BL_) bleaching. The insert picture shows a SEM image of the specimen surface at high magnification at T_BL_.

**Figure 3 polymers-13-02941-f003:**
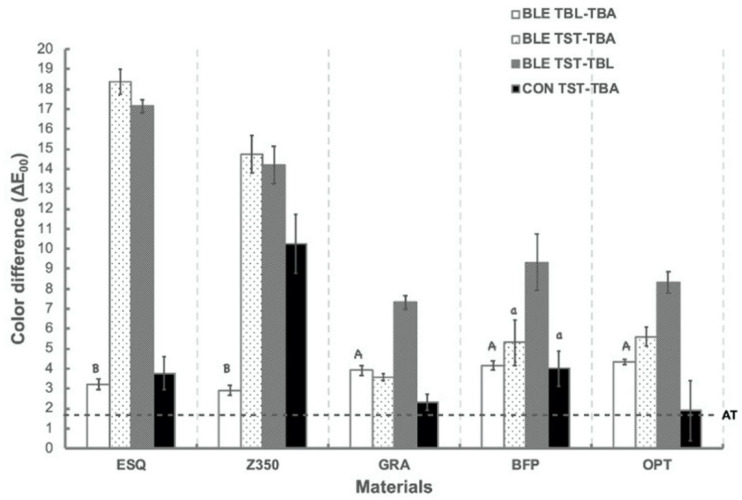
Mean and standard deviation values of color differences (ΔE_ab_*) between 2 different measuring time points (T_BL_-T_BA_, T_ST_-T_BA_, and T_ST_-T_BL_) for each material from the BLA and CON groups. Dashed line at 2.7 represents the acceptability threshold (AT) [8]. Same capital letter indicates no static difference between materials. Same lowercase letter indicates no static difference between time points for each material.

**Figure 4 polymers-13-02941-f004:**
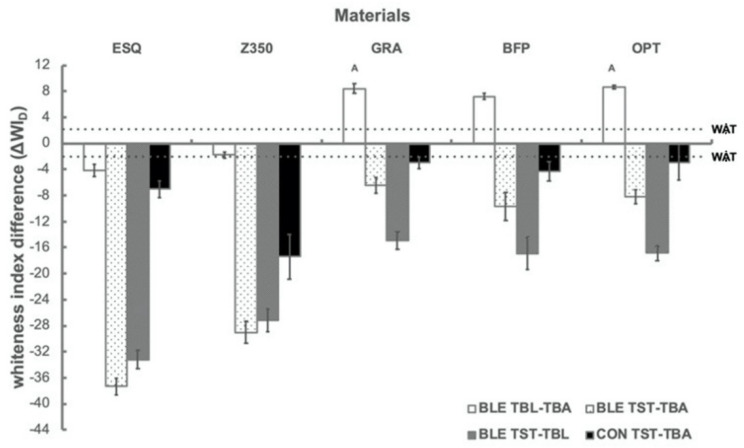
Mean and standard deviation values of variations in white index (ΔWI_D_) between 2 different measuring time points (T_BL_-T_BA_, T_ST_-T_BA_, and T_ST_-T_BL_) for each material from the BLA and CON groups. Dashed line at 2.9 represents the acceptability threshold (WAT) [16]. Same capital letter indicates no static difference between materials.

**Table 1 polymers-13-02941-t001:** Description of resin composites used in the study.

Material (Code)	Composition				Manufacturer
	Type	Organic Matrix	Fillers	Filler Amount	
Estelite Sigma Quick (ESQ)	Sub-micron	Bis-GMA, TEGDMA	monodispersing spherical SiO_2_ and ZrO_2_ (average 0.2 μm)	82 wt%	Tokuyama Dental, Tokyo, Japan
Filtek Z350XT (Z350)	Nano-filled	Bis-GMA, UDMA, Bis- EMA, TEGDMA, and PEGDMA	Non-agglomerated nano-particles of silica (20 nm), nano-agglomerates formed of zirconia/silica particles (0.6–1.4 μm)	72.5 wt%	3M ESPE, Monrovia, CA, USA
Grandio (GRA)	Nano-hybrid	Bis-GMA, Bis-EMA, TEGDMA	glass ceramic (particle size 1 μm), silicon dioxide nanoparticles (20–40 nm)	87 wt%	Voco GmbH, Cuxhaven, Germany
Bulk Fill Posterior (BFP)	Bluk-filled	AUDMA, AFM, DDDMA, UDMA	an aggregated zirconia/silica cluster (comprised of 20 nm silica and 4 to 11 nm zirconia particles) a ytterbium trifluoride filler consisting of agglomerate (100 nm)	76.5 wt%	3M ESPE, Monrovia, CA, USA
OptiComp LC (OPT)	Nano-hybrid	Bis-EMA, UDMA, TEGDMA	glass ceramic (particle size 0.7 μm), silica dioxide nanoparticles (20–40 nm), ytterbium trifluoride (100 nm)	79.5 wt%	PacDent, Brea, CA, USA

Information collected from manufacturers’ brochures. Bis-GMA, bisphenol A-glycidyl methacrylate; TEGDMA, triethylene glycol dimethacrylate; UDMA, urethane dimethacrylate; Bis-EMA, bisphenol A ethoxylated dimethacrylate; PEGDMA, polyethylene glycol dimethacrylate; AUDMA, a high molecular weight aromatic dimethacrylate; AFM, addition-fragmentation monomers; DDDMA, 1, 12-Dodecanediol dimethacrylate.

**Table 2 polymers-13-02941-t002:** Mean (standard deviation) of color parameters in CIELAB system and whiteness index (WI_D_) for specimens of each material in bleaching group before (T_BA_) and after bleaching procedure (T_BL_) and staining treatment (T_ST_) from BLE group.

		T_BA_	T_BL_	T_ST_
ESQ	L*	90.38 (0.18)	88.85 (0.37)	76.63 (0.75)
	a*	−6.45 (0.36)	−4.24 (0.47)	2.47 (0.45)
	b*	12.38 (0.59)	10.71(0.73)	20.63 (0.24)
	WI_D_	51.06 (0.83)	46.93 (1.42)	13.64 (1.58)
Z350	L*	82.92 (0.27)	81.97 (0.35)	71.21 (0.86)
	a*	−1.15 (0.10)	0.46 (0.12)	6.44 (0.46)
	b*	38.88 (0.56)	36.63 (0.26)	43.88 (0.47)
	WI_D_	5.50 (0.73)	3.70 (0.36)	−24.05 (2.05)
GRA	L*	62.94 (0.13)	65.75 (0.14)	60.33 (0.34)
	a*	−1.27 (0.10)	−3.53 (0.33)	−0.01 (0.58)
	b*	31.41 (0.33)	29.94 (0.42)	33.33 (0.76)
	WI_D_	3.00 (0.46)	11.43 (1.09)	−3.46 (1.17)
BFP	L*	64.25 (0.44)	66.73 (0.52)	60.49 (0.39)
	a*	−3.31 (0.10)	−4.34 (0.15)	−1.48 (0.16)
	b*	14.17 (0.28)	11.01 (0.12)	17.24 (1.78)
	WI_D_	27.44 (0.37)	34.66 (0.54)	17.75 (2.43)
OPT	L*	75.28 (0.43)	77.60 (0.42)	71.13 (0.47)
	a*	−3.90 (0.12)	−5.54 (0.17)	−0.42 (0.37)
	b*	24.71 (0.38)	21.47 (0.31)	23.04 (0.05)
	WI_D_	23.26 (0.13)	31.92 (0.31)	15.07 (0.86)

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
