# Peer review of "Color Stability and Staining Susceptibility of Direct Resin-Based Composites after Light-Activated In-Office Bleaching"

_polymers, 2021, doi:10.3390/polym13172941_

Round 1

Reviewer 1 Report

  • Abstract: please detail all abbreviations (such as BLE, CON)
  • Introduction - please cite similar studies mentioned (line 79)
  • Table 1 - did the investigators contact the PacDent for composition details? If the were and didn't respond, then this information needs to stated clearly. This is critical information given the obtained results;
  • Sample details and preparation (including polymerisation lightwave and any finishing/polishing) missing. Why were the composite resin samples light-cured for 80 seconds (long photo activation for more conversion, I think, but doesn't represent clinical recommendations)?
  • Why was a black background used, as opposed to 18% neutral grey?
  • Would storage in artificial saliva result in different findings?

Author Response

Dear Editor:

Please find the revised paper entitled “Color stability and staining susceptibility of direct resin-based composites after light-activated in-office bleaching” (Manuscript ID: polymers- 1358662) enclosed. We have incorporated all of the suggestions made by the reviewers. These changes are highlighted in red within the manuscript. Please see below, in red, for a point-by-point response to each reviewer’s comments and concerns.

Response to reviewer 1:

  1. Abstract: please detail all abbreviations (such as BLE, CON).

Response: We thank the reviewer for bringing up this point. The following sentences have been added in lines 24-25:

“Ten disc-shaped specimens of each RBC were fabricated and divided into bleaching (BLE) and non-bleaching (CON) groups (n=5 for each group).”

  1. Introduction - please cite similar studies mentioned (line 79)

Response: Thank you for pointing this out. We have inserted the reference accordingly on line 79.

  1. Table 1 - did the investigators contact the PacDent for composition details? If the were and didn't respond, then this information needs to stated clearly. This is critical information given the obtained results;

Response: Thank you for pointing this out. The information has been obtained after we proactively contacted with PacDent. We have updated the information in discussion section.

  1. Sample details and preparation (including polymerisation lightwave and any finishing/polishing) missing. Why were the composite resin samples light-cured for 80 seconds (long photo activation for more conversion, I think, but doesn't represent clinical recommendations)?

Response: The duration of photoactivation was set at 80 seconds due to the fully setting and conversion. We thank the reviewer for pointing out the necessity to add this information in lines 95-103:

“Each composite was packed into the mold, which was positioned over a polyester strip on a glass plate, and gently pressed with a transparent acrylic sheet on the top of the mold. The specimens were then light-cured with a light-emitting diode (LED) device with ap-proximately 800 mW/cm2 for 80 seconds (3M ESPE Dental Products; Monrovia, CA, USA) to ensure fully conversion of the photoactivation. Immediately, the specimens were polished using SiC #1000 sandpaper and diamond paste for 20 seconds, respectively. After rinsing and gently drying, the thickness of the specimens was then verified with a digital mi-crometer (MDC-250; Mitutoyo, Kawasaki, Japan), and stored in deionized water at 37 °C for 24 hours.”

  1. Why was a black background used, as opposed to 18% neutral grey?

Response: Thank you for pointing this out. In the present study, color measurements were carried out on a black background to reflect the real oral environment while evaluating the specimens' final colors. We will insert this sentence in lines 228-229.

  1. Would storage in artificial saliva result in different findings?

Response: Thank you for this suggestion. It would have been interesting to explore this aspect. However, in our study, only the deionized water was used. We have specified the limitations in the discussions section which need to be further investigated.

Reviewer 2 Report

Dear author,

The paper is good

Please prepare the suggestion from below:

1. Please include figures and tables at better resolution

2. Please remove figure 1

Author Response

Dear Editor:

Please find the revised paper entitled “Color stability and staining susceptibility of direct resin-based composites after light-activated in-office bleaching” (Manuscript ID: polymers- 1358662) enclosed. We have incorporated all of the suggestions made by the reviewers. These changes are highlighted in red within the manuscript. Please see below, in red, for a point-by-point response to each reviewer’s comments and concerns.

Response to reviewer 2:

  1. The paper is good.

Response:  We thank the reviewer for this encouraging comment.

  1. Please prepare the suggestion from below:
  • Please include figures and tables at better resolution

Response: Thank you for pointing this out. We have replaced 600 DPI resolution images.

  • Please remove figure 1

Response:  We have removed figure 1 and made the correction accordingly in the revised text.

Round 2

Reviewer 1 Report

Thank you for considering my feedback and comments. 

Author Response

 1. Thank you for considering my feedback and comments.

Response: We thank the reviewer for this encouraging comment.

This manuscript is a resubmission of an earlier submission. The following is a list of the peer review reports and author responses from that submission.

Round 1

Reviewer 1 Report

While the work seems to be a useful statement of applied research results, unfortunately from the scientific point of view manuscript is unacceptable. The presented research results take the form of a report on the analysis of color changes without an analysis of the structure of the investigated commercial materials, in which case relying on literature references is unfortunately insufficient. Unfortunately, the novelty of the described work is also insufficient, in the literature you can find many similar works, including those taking into account the impact of 35% HP bleaching before immersion in red wine. Due to the above, I cannot recommend the article for publication.

Reviewer 2 Report

I have reviewed the manuscript titled “Color stability and staining susceptibility of direct resin-based composites after light-activated in-office bleaching” submitted to “Polymers” for publication. In this study, authors have evaluated the color stability and staining susceptibility of five direct resin-based composites subjected to light-activated in-office bleaching with 40% hydrogen peroxide.

The data presented is insufficient and further variables can be added to enhance the impact of the study. For example, study design can be improved by adding more bleaching technique effects, adding more variables in addition to red wine. Further experimentation can support the presented findings.

The discussion is very superficial and lack in depth analysis on results and findings;